# PRAME Is a Novel Target of Tumor-Intrinsic Gas6/Axl Activation and Promotes Cancer Cell Invasion in Hepatocellular Carcinoma

**DOI:** 10.3390/cancers15092415

**Published:** 2023-04-22

**Authors:** Viola Hedrich, Kristina Breitenecker, Gregor Ortmayr, Franziska Pupp, Heidemarie Huber, Doris Chen, Sarthak Sahoo, Mohit Kumar Jolly, Wolfgang Mikulits

**Affiliations:** 1Center for Cancer Research, Comprehensive Cancer Center, Medical University of Vienna, 1090 Vienna, Austria; viola.hedrich@meduniwien.ac.at (V.H.);; 2Department of Chromosome Biology, Max Perutz Labs Vienna, University of Vienna, 1030 Vienna, Austria; 3Centre for BioSystems Science and Engineering, Indian Institute of Science, Bangalore 560012, India

**Keywords:** HCC, Axl, Gas6, PRAME, EMT, invasion, dedifferentiation, immunotherapy

## Abstract

**Simple Summary:**

The receptor tyrosine kinase Axl is upregulated in up to 40% of hepatocellular carcinoma (HCC) cases correlating with an unfavorable prognosis. It is an open issue how Axl and its ligand Gas6 drive disease development at a molecular level. The aim of this study was to identify target genes of Gas6/Axl and to assess their contribution to HCC progression. One of the targets is the cancer-testis antigen PRAME (preferentially expressed antigen in melanoma), which is currently exploited for its capacity in T cell-based immunotherapy. We show that PRAME induces hepatic dedifferentiation, epithelial-to-mesenchymal transition and invasiveness in liver cancer cells. Together, these data provide evidence that PRAME is induced by the Gas6/Axl/Mek/Erk1/2 signaling axis and exerts pro-oncogenic functions in HCC.

**Abstract:**

(1) Background: Activation of the receptor tyrosine kinase Axl by Gas6 fosters oncogenic effects in hepatocellular carcinoma (HCC), associating with increased mortality of patients. The impact of Gas6/Axl signaling on the induction of individual target genes in HCC and its consequences is an open issue. (2) Methods: RNA-seq analysis of Gas6-stimulated Axl-proficient or Axl-deficient HCC cells was used to identify Gas6/Axl targets. Gain- and loss-of-function studies as well as proteomics were employed to characterize the role of PRAME (preferentially expressed antigen in melanoma). Expression of Axl/PRAME was assessed in publicly available HCC patient datasets and in 133 HCC cases. (3) Results: Exploitation of well-characterized HCC models expressing Axl or devoid of Axl allowed the identification of target genes including PRAME. Intervention with Axl signaling or MAPK/ERK1/2 resulted in reduced PRAME expression. PRAME levels were associated with a mesenchymal-like phenotype augmenting 2D cell migration and 3D cell invasion. Interactions with pro-oncogenic proteins such as CCAR1 suggested further tumor-promoting functions of PRAME in HCC. Moreover, PRAME showed elevated expression in Axl-stratified HCC patients, which correlates with vascular invasion and lowered patient survival. (4) Conclusions: PRAME is a bona fide target of Gas6/Axl/ERK signaling linked to EMT and cancer cell invasion in HCC.

## 1. Introduction

Hepatocellular carcinoma (HCC) is a devastating disease accounting for 75–90% of liver cancer patients [1,2]. It is frequently diagnosed at advanced stages due to lack of symptoms and insufficient screening of patients with premalignant chronic liver diseases, resulting in limited therapeutic options [3,4]. As HCC is a heterogeneous disease at the molecular and cellular level, these treatment options have modest effects on patient survival [5]. In order to develop patient-tailored therapy strategies to better meet the needs of those affected, a more detailed understanding of the molecular mechanisms underlying HCC is required.

Upregulation of the receptor tyrosine kinase Axl, a member of the TAM receptor family together with Tyro3 and MerTK, associates with poor prognosis of HCC patients [6]. Nuclear serine/threonine kinases NUAK1/2, signal transducer and activator of transcription 3, yes-associated protein 1 and frizzled-2 are capable of inducing Axl expression in about 40% of HCC cases [6,7,8,9]. Axl can signal via MAPK, PI3K/Akt or Src among others favoring cell survival, proliferation, angiogenesis, epithelial-to-mesenchymal transition (EMT), cell invasion and drug resistance [10]. HCC cells frequently upregulate the expression of the EMT transcription factors (EMT-TFs) Snail, Slug, Twist and Zeb1/2 resulting in reprogramming to mesenchymal gene expression [7,11,12]. Axl is an important mediator of EMT in various malignancies associating with the expression of genes regulating stemness and metastasis [13,14]. In HCC, it is poorly understood how the activation of Axl selectively induces a gene expression program driving tumor progression and metastasis.

PRAME (preferentially expressed antigen in melanoma) has been identified as an antigen recognized by cytotoxic T lymphocytes, which is currently exploited for CAR-T cell therapy [15,16,17,18]. High expression of PRAME is restricted to testis in adult tissues, while it is frequently augmented in malignant settings including melanoma or lung cancer, which adds PRAME to the family of cancer-testis antigens (CTAs) [19,20]. Interestingly, tumor-promoting functions were identified for some CTAs including e.g., HORMAD1 and NANOS3, which foster lung cancer invasion by EMT [21,22]. PRAME represses retinoic acid receptor (RAR) signaling by binding to RAR and hampering retinoic acid (RA)-induced receptor activation and target gene transcription [16]. Thereby PRAME rescues melanoma cells from RA-mediated anti-proliferative effects [16]. Additional functions of PRAME are suggested as reduced PRAME expression interferes with melanoma cell proliferation even in the absence of RA [16]. In accordance, PRAME expression is significantly upregulated in breast cancer where it favors EMT and immunosuppression [23,24]. In a cellular model of laryngeal squamous cell carcinoma, PRAME promotes proliferation, EMT and invasion [25].

In this study, we identified target genes of Gas6/Axl signaling, which revealed PRAME as an oncogenic driver in HCC cells. PRAME expression induced by Gas6/Axl/MAPK promotes EMT-associated gene expression, cell motility and RA-independent interaction with nuclear proteins, suggesting a novel molecular link between Gas6/Axl signaling and PRAME.

## 2. Materials and Methods

### 2.1. Cell Culture

The human liver cancer cell lines HLF, HLE cells were propagated in DMEM supplemented with 10% fetal calf serum (FCS) while SNU475 were cultured in RPMI-1640 plus 10% FCS. HepG2 as well as Hep3B cells received EMEM plus 10% FCS. HepG2, Hep3B and SNU475 were purchased from ATCC (Manassas, VA, USA), while HLF and HLE cells were kindly provided by Dr. Steven Dooley (University of Heidelberg, Germany). Axl signaling was induced by administration of 500 ng/mL recombinant Gas6 (R&D Systems, Minneapolis, MN, USA) or by Axl-agonistic antibody AF154 (R&D Systems, Minneapolis, MN, USA). Inhibition of Axl was performed by 1–10 µM Cabozantinib (Eubio, Atlanta, GA, USA) by incubation between 15 min to 48 h [26,27]. The Mek inhibitors Trametinib (SelleckChem, Houston, TX, USA) and U0126 (Thermofisher, Waltham, MA, USA) were used to interfere with MAPK signaling. KX-391 (SelleckChem, Houston, TX, USA) was employed to inhibit Src. HepG2 and Hep3B cells were lentivirally transfected with particles containing the pWPI vectors with green fluorescent protein (GFP) or PRAME-GFP. GFP-positive cells were enriched via cell sorting. The Axl locus was disrupted as described previously [28]. For generation of PRAME-knockout (KO) cell lines, small guide (sg)RNAs were cloned into the pSpCas9(BB)-2A-Puro V2.0 vector (Addgene, Watertown, MA, USA) and hepatoma cells were transfected with the plasmids using Lipofectamine 2000 (Thermofisher, Waltham, MA, USA) as described [29]. Cells were selected using 1.5 mg/mL (HLF) or 1 mg/mL (HLE) puromycin (Merck, Darmstadt, Germany) for 48 h. Surviving cells were expanded and clonal lines were isolated and propagated. All cells were propagated under standard cell culture conditions of 37 °C and 5% CO_2._ Cell line identity was validated via short tandem repeat analysis and the absence of mycoplasma was ensured by routine testing.

### 2.2. Knockdown by RNA Interference

Interference studies were performed as described recently [6,30]. In brief, cells were seeded in 96-well, 12-well or 6-well plates and either transfected with 50 nM of on-target^plus^ non-targeting pool small interfering (si)RNA or with 50 nM on-target^plus^ SMARTpool siRNA targeting human PRAME (Dharmacon, Lafayette, CO, USA) using Oligofectamine (Thermofisher, Waltham, MA, USA) according to the manufacturer’s protocol. Cells were cultivated in presence of siRNA between 24 and 96 h and further processed.

### 2.3. 2D Wound Healing Assay

7.5 × 10^5^ cells were seeded in 6-well plates to study two-dimensional (2D) cell migration. Cell layers were scratched with pipette tips to introduce artificial wounds. Plates were washed to remove detached cells and images were immediately taken by a phase contrast microscope (Nikon, Tokyo, Japan) and 48 h after scratching. Quantification of the migrated cells was performed by ImageJ software (https://imagej.nih.gov/ij/). Three independent experiments were conducted for each analysis.

### 2.4. 3D Invasion Assay

For studying three-dimensional (3D) cell invasion, the protocol from Osswald et al. was adapted [31]. In brief, 1 × 10^3^ cells were seeded in cell-repellent U-bottom 96-well plates (Greiner Bio-One, Kremsmünster, Austria) for 72 h to form spheres. Spheres were embedded in a 1.5 mg/mL rat-tail collagen I solution supplemented with 0.125% sodium bicarbonate, 15 mM HEPES and 2% FCS. Polymerization was induced by adding NaOH and incubation at 37 °C in 5% CO_2_ for 1.5 h. Collagen gels were overlayed with 100 µL culture medium and images were taken by phase contrast microscopy (Nikon, Tokyo, Japan) 24 and 48 h post embedding. Quantification of invaded cells into the gel was performed by ImageJ software (https://imagej.nih.gov/ij/) [32]. Three independent experiments were conducted for each analysis.

### 2.5. Clonogenic Survival Assay

1 × 10^3^ cells were seeded in 6-well plates and incubated for 10 days at 37 °C and 5% CO_2_. Colonies were fixed with methanol/acetic acid (3:1) (Lactan, Graz, Austria/VWR, Radnor, PA, USA), stained with 0.25% crystal violet (Merck, Darmstadt, Germany) and counted. Three independent experiments were conducted for each analysis.

### 2.6. Proliferation Kinetics

1 × 10^4^ cells were seeded in 12-well plates and cell numbers were determined every second to third day for up to 144 h using the cell analyzer CASY (Schärfe Systems, Reutlingen, Germany). Three independent approaches were conducted for each analysis.

### 2.7. Transcriptome Profiling by RNA-seq

HLF, HLF-nickase, HLF-Axl-KO and HLF-Axl-KO-wt-Axl cells were stimulated with 500 ng/mL Gas6 and total RNA was isolated using the Monarch total RNA miniprep kit (New England Biolabs, Ipswich, MA, USA) in three independent experiments. Isolated RNA was quantified via Nanodrop (Thermofisher, Waltham, MA, USA) and the quality was checked. Each RNA sample was divided into three aliquots: one for QuantSeq 3′ mRNA-seq, one for validation of sequencing results and one as a backup which was snap frozen and stored at −80 °C. Library preparation and sequencing were performed by Lexogen (Greenland, NH, USA). In brief, quality controlled and quantified RNA was used to prepare a QuantSeq 3′ mRNA-seq (fwd) library suitable for Illumina sequencing, containing Unique Molecular Identifiers (UMIs). Library preparation was followed by another quality control. The QuantSeq 3′ (fwd) library was subsequently subjected to SR76 high output QuantSeq 3′ mRNA-sequencing via Illumina NextSeq 500, followed by lane mix quality control and demultiplexing. Spike-In RNA Variant Controls (SIRVs)—which are synthetic, unique RNA transcripts—were included as controls for RNA-sequencing.

If not stated otherwise, analyses were performed with the help of custom R (v. 4.0.0) scripts and RStudio. Trimming of poly-As from the 3′-ends of the 76 bp long reads was performed by cutadapt (v. 1.12; with options -O 3 -e 0.10 -m 20 --max-n 2), retaining reads with at least 20 bp. Alignments to the human reference genome GRCh38.p13 (GCA_000001405.28) and the corresponding Gencode gene annotation Release 33 were conducted using STAR 2.7.3a [33], with up to 1000 hits, a maximum error rate of 5% and counts per gene as output.

For visualizing potential batch effects, normalized samples were clustered hierarchically (R hclust, stats package), or analyzed by principal component analysis and pairwise plotting of the first five principal components (PCA, package FactoMineR).

For pair-wise differential expression analyses, DESeq2 [34] was used. Genes with less than three counts in more than three samples were removed. Moreover, duplicate gene symbols were filtered based on expression level, resulting in a final set of 15,063 genes. Default parameters were selected except for the dispersion model fitting, which was chosen to be local. Since PCA showed that the preparation date contributed to the sample variance, preparation dates were included in the design formula. To obtain batch-corrected results, limma batch correction was performed. For visualization of significantly differentially expressed genes in heat maps, expression values were furthermore log-vst transformed. All reported p-values, including cut-offs, refer to (Benjamini–Hochberg) multiple-test corrected q-values. 

The RNA-seq data are available at GEO database under GSE229731.

Gene set enrichment analyses were performed with the command-line version 4.1.0 of the UCSD/BROAD GSEA tool [35]. Signal2Noise ranking as well as gene set permutation were selected, when more than six samples existed per group, otherwise phenotype permutation was performed. Gene sets Hallmark (H), BioCarta, KEGG, PID, Reactome (CP), and GO (C5) were downloaded from the Molecular Signatures Database (MSigDB) and filtered for a set size of 30–500 genes were queried.

Venn diagrams were generated with the help of Ghent University online resources available at Bioinformatics and Evolutionary Genomics (Van de Peer Lab; https://bioinformatics.psb.ugent.be/webtools/Venn/). Axl targets were defined as differentially expressed genes between Axl-proficient and Axl-deficient cells. Comparisons between (i) parental and CRISPR control cell line (only transfected with enzyme but no sgRNAs) and (ii) parental and rescue cell line served as controls. Significantly differentially expressed genes (*p*-value < 0.05 and absolute log2-fold change > +1.0) of selected pairwise comparisons were used to generate Venn diagrams.

### 2.8. Reverse Transcription, Quantitative Polymerase Chain Reaction (RT-qPCR)

Total RNA was isolated using Monarch total RNA miniprep kit (New England Biolabs, Ipswich, MA, USA) according to the manufacturer’s protocol. Isolated RNA was quantified by Nanodrop (Thermofisher, Waltham, MA, USA) and cDNA was generated by reverse transcription via the iScript cDNA synthesis kit (Biorad, Neuried, Germany). cDNA was subsequently used as an input for qPCR applying the Luna universal qPCR master mix (New England Biolabs, Ipswich, MA, USA). qPCR was performed on the CFX connect real-time PCR system (Biorad, Neuried, Germany) and quantified via the CFX Maestro Software version 4.0.2325.0418 (Biorad, Neuried, Germany). The following primers were used: PRAME: forward: 5′-GTGACCTGTGAACAGCAACT-3′, reverse: 5′-CCACGCACGTCTGAGAGTAAT-3′; RPL41: forward: 5′-CAAGTGGAGGAAGAAGCGA-3′, reverse: 5′-TTACTTGGACCTCTGCCTC-3′; TWIST: forward: 5′-TTCTCGGTCTGGAGGATGGA-3′, reverse: 5′-AATGACATCTAGGTCTCCGGC-3′; SNAI1: forward: 5′-GTAATGGCTGTCACTTGTCG-3′, reverse: 5′-TGTAAACATCTTCCTCCCAGG-3′.

### 2.9. Western Blotting

Immunoblotting was performed as described previously [30]. The primary antibodies were used at a dilution of 1:1000 if not indicated otherwise: anti-phospho-Axl (#5724), anti-phospho-Erk1/2 (#9101), anti-Erk1/2 (#4695), anti-phospho-Src (#6943) and anti-Src (#2109) were purchased from Cell Signaling (Danvers, MA, USA). Anti-Axl (#AF154) and anti-PRAME (#ab219650) were obtained from R&D Systems (Minneapolis, MN, USA) and Abcam (Cambridge, England, UK), respectively. Anti-actin (#A2206, Merck, Darmstadt, Germany) was applied at 1:5000 as a loading control. Horseradish peroxidase-conjugated secondary antibodies (Vector Laboratories, Newark, CA, USA) were diluted 1:10.000. Abundance of proteins was quantified via densitometry using ImageJ software (https://imagej.nih.gov/ij/). Peak areas were quantified and normalized to the actin loading control. For the control versus the treatment experiments, the abundance of proteins upon interference was set relative to the control.

### 2.10. Immunoprecipitation (IP)

500 µg of HLF lysate was mixed with anti-PRAME antibody (#ab219650, Abcam, Cambridge, England, UK) as indicated by the manufacturer and incubated overnight at 4 °C. Subsequently, 50 μL of protein G-coated magnetic beads (Mag Sepharose™ Xtra; Merck, Darmstadt, Germany) was added and incubated for 1 h at 4 °C. The IP complexes were washed with PBS. For mass spectrometry, samples were delivered “on-beads” on ice and immediately processed. For immunoblotting, proteins were eluted in 30 μL sodium dodecyl sulfate (SDS) polyacrylamide gel electrophoresis sample buffer containing 100 mM Tris/HCl pH 7.4, 5% SDS, 5 mM DTT and 70 mM β-mercaptoethanol. Samples were either analyzed immediately or stored at −20 °C.

### 2.11. Mass Spectrometry Analyses of IPs

IP proteomics were performed by the Mass Spectrometry Facility of the Max Perutz Labs (MPL, Vienna, Austria) using the Vienna Biocenter Core Facility instrument pool. For data processing, a database search was performed with MaxQuant 2.1.4.0. Search parameters: standard, trypsin/P, with oxidation(M), protein N-term acetylation as variable and carbamidomethylation (C) as fixed modification. All data were filtered at 1% PSM + protein + site false discovery rate and reverse hits were removed. Missing label-free quantification values were imputed into log space by drawing random values from a normal distribution (estimated from sample label-free quantification intensity distribution and shifted by −1.8 standard deviations with a width of 0.3 standard deviations) after filtering out contaminants and proteins with less than two razors and unique peptides. For pairwise comparison, the log_2_ fold change and mean label-free quantification were calculated and plotted against each other (MAplot).

### 2.12. Correlation Analysis

To plot the correlation heat map of PRAME-associated transcription factors, the top correlated transcription factors based on the comparison of their RNA expression levels with that of PRAME were identified. Next, the correlations were calculated for liver cancer cell lines based on the Cancer Cell Line Encyclopaedia (CCLE) data and only those transcription factors were considered whose Spearman correlations coefficient was either greater than 0.55 or lesser than −0.55. This yielded a list of 35 positively correlated and 5 negatively correlated transcription factors. A gene–gene correlation matrix with gene-wise hierarchical clustering of these 40 genes and PRAME was plotted to highlight the positive and negative gene-gene correlations. A comparable mode was used to generate a gene-gene correlation matrix with gene-wise hierarchical clustering of PRAME and its interaction partners identified by IP-mass spectrometry. Spearman correlation coefficient was calculated to construct the correlation matrix.

### 2.13. In Silico Analysis

The Cancer Genome Atlas (TCGA) liver cancer (LIHC) data were assessed via the open-source resources cBioPortal and Xenabrowser for analysis and visualization [36,37,38]. Sample information regarding sample type (“solid tissue normal”, “primary tumor” and “recurrent tumor”) and histological grade (G1-G4) was obtained via Xenabrowser. Additionally, transcript expression of *AXL*, *PRAME*, *TWIST* and *SNAI1* was retrieved in log_2_(fpkm-uq + 1). PRAME expression was plotted against sample type, histological grade or in *AXL*-stratified samples (*AXL*^high^ = 75% quartile, *AXL*^low^ = 25% quartile). cBioPortal allowing a more general approach was used to stratify total gene expression of TCGA LIHC data depending on *PRAME* expression to identify overall differentially expressed transcripts between *PRAME*^high^ (=75% quartile) and *PRAME*^low^ (=25% quartile) samples. Furthermore, *PRAME* expression was correlated with histological grade and occurrence of vascular invasion. Kaplan–Meier plots were generated applying the Kaplan–Meier plotter (https://kmplot.com/analysis/) [39]. For network analysis, the STRING database was used (https://string-db.org/).

### 2.14. Immunohistochemistry of Primary HCC Patient Samples

PRAME expression was assessed by immunohistochemical staining of 4 µm thick paraffin-embedded tumor samples collected from 133 HCC patients with anti-PRAME antibody (#ab219650, Abcam, Cambridge, UK). All patients received orthotopic liver transplantation at the Department of Transplantational Surgery, Medical University of Vienna, between 1982 and 2002 as described previously [6,30]. Two individual certified pathologists performed grading and histological assessment. Samples were present in triplicates from each case on tissue microarray slides. Anti-Axl staining (#AF154, R&D Systems, Minneapolis, MN, USA) was performed previously in this cohort [6]. PRAME assessment was performed in this study [6]. In brief, anti-PRAME antibody was used at a dilution of 1:500 and incubated overnight at 4 °C. Secondary antibodies were applied as recommended by the manufacturer’s protocol (Vector Laboratories, USA). The Vectrastain ABC kit (Vector Laboratories, Newark, CA, USA) was employed using diaminobenzidine (“DAB”) as a substrate for visualization. Counterstaining was performed with Gill’s hematoxylin solution No. 3 (Carl Roth, Karlsruhe, Germany). Tissue Studio v4.4.1 image analysis software (Definiens, Munich, Germany) was applied for the assessment of immunohistochemical staining. Samples were grouped depending on positive or negative PRAME expression and correlated with available clinical data.

### 2.15. Statistical Analysis

Data were expressed as means ± standard deviation (SD). Statistical significance was assessed by unpaired Student’s *t*-test for continuous data. For categorical data, Fisher’s exact test was employed. Log-rank tests were performed for the analysis of Kaplan–Meier survival curves using GraphPad Prism v5.01 (GraphPad Software Inc., San Diego, CA, USA). Different *p*-values were considered as statistically significant: * *p* < 0.05, ** *p* < 0.01, *** *p* < 0.001.

## 3. Results

### 3.1. Identification of Gas6/Axl-Dependent Targets in HCC Cells

To assess target genes of Gas6/Axl signaling, which regulates tumor-intrinsic gene expression [7,40], we examined mesenchymal-like HCC cells expressing Axl (wild type; wt) versus CRISPR/Cas9-mediated Axl knockout cells (KO, Axl^−^) versus Axl-KO cells harboring reconstituted Axl expression (Axl^−^ wt) by RNA-seq (Figure 1A). Cells expressing nickase only (nick) were used as the control. Pairwise comparisons were used to generate VENN relations which revealed AGPAT3, ALDH1A1, C15orf48, CNN1, LITAF, MAT1A, NOL4L and PRAME as putative targets of Gas6/Axl signaling (Figure 1A,B; Appendix A).

We further focused on PRAME as it exhibited a 5.4 to 6.0-fold upregulation in the various experimental settings (Figure 1B; left, middle and right panel) and was recently linked to an unfavorable prognosis in HCC [41]. Accordingly, differential expression of PRAME was confirmed on transcript and protein level, demonstrating a low abundance of PRAME in Axl-deficient cells (Figure 1C,D). The modulation of Axl activation was validated by administration of Gas6 in the absence or presence of the Axl inhibitor Cabozantinib or by treatment with a stimulatory Axl antibody (Figure 1E) [26,27]. In this line, PRAME expression was approximately 30% reduced upon interference with Cabozantinib when compared to HCC cells stimulated with Gas6 or Axl antibody. Taken together, these data indicate that PRAME expression depends on Gas6/Axl signaling in HCC cells and can be modulated by genetic or pharmacological intervention.

### 3.2. Expression of PRAME Correlates with Dedifferentiation of HCC Cells and EMT

Gene Set Enrichment Analysis (GSEA) based on RNA-seq data identified Axl-regulated biological processes including tissue remodeling, main axon and LEF1_UP,V1_DN signature, which correlates with EMT (Appendix A) [12,42,43,44,45].

As CTAs have been linked to tumor progression by modulating epithelial cell plasticity [21,22,23,24,25,46], we analyzed whether PRAME contributes to an EMT phenotype in liver cancer. Therefore, we stratified publicly available HCC patient datasets according to PRAME expression and identified transcripts upregulated in PRAME-high patients including *CTNND2*, *DLK1*, *KRT19*, *EPCAM*, *DKK1*, *AFP*, *MMP7* (Figure 2A). Notably, the EMT-TFs *SNAI1* and *TWIST* were upregulated in HCC patients exhibiting high expression of PRAME (Figure 2A,B). Network analysis of upregulated genes revealed their enrichment in signatures of cancer cell growth and fetal liver (e.g., *AFP*, *DLK*, *EPCAM* and *KRT19*) (Appendix A). Importantly, onco-fetal reprogramming was recently described as a key feature of HCC [47]. In contrast, CYP enzymes such as *CYP1A2*, *CYP2A7*, *CYP3A4*, *CYP8B1* and the androgen receptor (*AR*) were downregulated (Figure 2A). The transcripts correlating with low abundance in PRAME-high samples are linked to normal liver function based on network analysis (Appendix A).

Comparably, analysis of available human liver cancer cell lines revealed a strong positive correlation between PRAME and the expression of *TWIST* or *LEF1* (Appendix A). The latter was among the enriched gene sets in Axl-expressing cells associating with EMT (Appendix A) [44,45]. In contrast, *PRAME* negatively correlated with *FOXA2* and *AR* expression in human liver cancer cell lines (Appendix A). Both are regulators of hepatocyte differentiation [48,49,50]. In accordance, analysis of liver cancer cell line profiles revealed negative correlations between *PRAME* and androgen response as well as adult hepatocyte signatures (Appendix A). Together these findings from in silico analysis link PRAME to loss of liver function and HCC-related onco-fetal reprogramming [47,48].

To validate these observations in cellular models, we assessed PRAME and Axl expression in different HCC cell lines. Accordingly, Axl-proficient HLF, SNU475 and HLE cells, associating with a mesenchymal-like phenotype [6], expressed high levels of PRAME, while the epithelial liver cancer cell lines HepG2 and Hep3B lacked expression of Axl and PRAME (Appendix A). Therefore, we generated epithelial cells exogenously expressing PRAME such as HepG2-PRAME and Hep3B-PRAME, as well as HLF and HLE PRAME-KO cell lines (Appendix A). Noteworthy, no morphological changes were obvious upon modulation of PRAME expression. Next, we examined the expression of *SNAI1* and *TWIST* in those cell lines. Notably, PRAME expression in epithelial HCC cells increased *SNAI1* and *TWIST* levels, whereas PRAME-KO in mesenchymal HCC cells reduced their abundance up to 50% (Figure 2C). These findings are in line with results obtained from the in silico analysis of patient and cell line expression profiles (Figure 2A,B and Appendix A). Therefore, we concluded that PRAME is crucially involved in establishing and maintaining an EMT-like phenotype during HCC progression.

### 3.3. PRAME Augments 2D Cell Migration and 3D Invasion

EMT is a fundamental process involved in cell migration, invasion and further aspects of metastasis [12,51]. As PRAME expression is linked to hepatocellular dedifferentiation and upregulation of EMT-TFs, we focused on the impact of PRAME on cell proliferation, migration and invasion by performing gain- and loss-of-function studies. The KO of PRAME reduced proliferation of HLF and HLE cells compared to the control (Appendix A). In a 2-dimensional (2D) setting, using wound healing assays, both PRAME-KD and PRAME-KO reduced the migratory potential while exogenous expression of PRAME in epithelial HCC cells elevated it by about 20% (Figure 3A, Appendix A). Stronger effects were obtained in a 3D setting by inoculating hepatospheres into collagen gels as both PRAME-KD and PRAME-KO decreased the invasive phenotype by more than 2-fold and increased the one in HepG2-PRAME more than 3.5-fold (Figure 3B and S4B). In addition, exogenous PRAME expression conferred a 30-fold increase in clonogenicity compared to GFP control (Figure 3C). In conclusion, these data provide evidence that PRAME is tumor-promoting because it positively influences proliferation and clonogenicity. In addition, high PRAME expression favors 2D migration and 3D invasion in HCC cells.

### 3.4. Axl-Induced MAPK Signaling Modulates PRAME Expression

Gas6-activated Axl can transduce a plethora of signals including the activation of Src, MAPK, PLCγ, Stat1 or AKT [10]. To assess which Axl-activated pathways affect PRAME expression, we stimulated Axl-proficient and Axl-deficient HCC cells with Gas6. Notably, we observed that phosphorylation of Src remained unaffected (Appendix A). In line with unaffected p-Src levels after Gas6 stimulation, reduced Src phosphorylation by interference with Src signaling via KX-391 failed to show changes in PRAME expression (Appendix A). However, phosphorylation of ERK1/2 was increased in Axl-expressing HLF and SNU475 cells upon stimulation with Gas6 (Figure 4A,B). 

Accordingly, interference with MAPK signaling by administration of either Trametinib or U0126 significantly reduced p-ERK1/2 levels (Figure 4C,D). Interestingly, PRAME expression remained unaffected at early time points, whereas inhibition of MEK/ERK1/2 for 24 to 48 h significantly reduced PRAME expression (Figure 4C,D). These data suggest that expression of pro-oncogenic PRAME depends on MAPK activation in Gas6/Axl-expressing HCC cells.

### 3.5. Identification of PRAME Binding Partners

As PRAME is described to interact with a variety of proteins [16,23,52,53,54,55,56], we aimed to identify binding partners of PRAME in HCC cells. First, we validated the immunoprecipitation (IP) capacity of the anti-PRAME antibody (Appendix A) and subsequently performed IP-mass spectrometry (IP-MS). Principle component analysis of IP/MS data showed clear antibody-dependent clustering of the samples, indicating successful IP/MS (Appendix A). Based on IP/MS, we enriched 41 proteins in the presence of anti-PRAME antibody including PRAME (Figure 5A and Appendix A). According to STRING database analysis, around 25% of all identified hits were classified as transcriptional repressors (Appendix A), suggesting that PRAME, together with its interaction partners, acts as a transcriptional regulator, which is in line with previous reports [16,53].

To address whether these proteins are potentially common interaction partners, we correlated the abundances of IP/MS-identified PRAME binding partners with PRAME levels in a large collection of liver cancer cell lines (Figure 5B). AGAP1, CCAR1 and SH3PXD2B were among those hits that exhibit a positive association with PRAME in the majority of assessed cellular models. Furthermore, we selected AGAP1, CCAR1 and SH3PXD2B for subsequent analysis as their expression was significantly associated with poorer survival probability in HCC (Figure 5C). High levels of CCAR1 correlated most significantly with worse outcome (Figure 5C, middle). Notably, CCAR1 belonged to the 25% of repressors based on network analysis and could be validated as an interaction partner of PRAME via IP and immunoblotting (Figure 5D and Appendix A).

Interestingly, all three interaction partners exhibited a significant positive correlation with PRAME expression in primary HCC patient samples (Figure 5E) while no correlation was found in normal tissue (Appendix A). In this line, AGAP1, CCAR1 and SH3PXD2B were expressed significantly higher in tumors compared to normal tissues (Appendix A) and their expression had a tendency to increase with histological grading (Appendix A) Together, these findings suggest that the interaction of PRAME with pro-oncogenic AGAP1, CCAR1 and SH3PXD2B is involved in establishing PRAME-dependent tumor-promoting functions.

### 3.6. Expression of PRAME and Axl in HCC Patients Correlates with Advanced Stage, Vascular Invasion and Poor Survival

To translate data from cellular models into HCC patients, we studied transcript levels of PRAME in publicly available liver cancer datasets focusing on its expression in normal tissue versus HCC, as well as PRAME expression in the context of histological grade and invasive phenotype. In accordance, PRAME expression was significantly elevated in HCC compared to adjacent tissue (Figure 6A) and progressively increased from low grade G1 to high grade G4 (Figure 6B,C left panel), indicating higher PRAME expression in increasingly dedifferentiated tumors compared to low grade, well differentiated ones. In addition, higher levels of PRAME correlated with micro- and macro-vascular invasion (Figure 6C right panel).

In line with the in vitro findings, we found PRAME transcripts were significantly higher expressed in Axl-stratified HCC cases (Figure 6D), and high levels of PRAME plus Axl associate with lower survival probability (Figure 6E). To confirm results obtained by the analysis of publicly available data based on transcript abundance, protein levels of PRAME were immunohistochemically determined in tissues of 133 HCC patients (Figure 6F), in which Axl expression was examined recently [6]. About 57% of PRAME-negative HCC samples lacked Axl, while 54% of PRAME-positive patients were Axl-positive as well (Figure 6G), which exhibited a more advanced tumor stage (Figure 6H) and a higher incidence of vascular invasion and recurrence (Figure 6I). Notably, PRAME and Axl expression shows a trend toward poorer survival in this cohort of HCC patients (Figure 6J). In conclusion, these data suggest that expression of PRAME is linked to Axl in HCC patients, favoring tumor progression as indicated by the correlation with advanced stages, increased invasion and a poorer survival probability of HCC patients.

## 4. Discussion

We identified eight target genes of Gas6/Axl signaling including the CTA PRAME in cellular models of HCC by applying RNA-seq and filtering differentially expressed genes for specific signals by removing background genes. In publicly available HCC patient data, high PRAME expression was linked to an increase in transcripts associated with EMT and dedifferentiation as well as to a decrease in hepatocyte differentiation. PRAME expression fostered the migratory and invasive potential of various liver cancer cell lines. We found that PRAME expression depends on Gas6/Axl/MAPK signaling and identified pro-oncogenic CCAR1, AGAP1 and SH3PXD2B as PRAME interaction partners. In HCC patient data, PRAME is positively associated with Axl expression, advanced HCC stages and poor prognosis.

We particularly focused on PRAME as its expression associates with EMT in breast cancer and laryngeal squamous cell carcinoma [23,24,25]. Furthermore, the CTA is linked to an unfavorable prognosis in HCC as well as an increased risk for hepatic metastasis and hepatic recurrence in gastric cancer [41,57]. PRAME correlates with a poorer outcome and was shown to confer growth advantage to liver cancer cell lines by inducing resistance to apoptosis [41,58]. These data indicate a link between PRAME and the induction of tumor-progressive characteristics. The analysis of PRAME-stratified HCC patient data showed loss of liver identity and induction of EMT markers such as *TWIST*, which was recapitulated in expression profiles of liver cancer cell lines and gain- and loss-of-function models. In this line, onco-fetal reprogramming was recently described as a key feature of HCC [47] and PRAME was shown to augment expression of EMT markers in triple negative breast cancer favoring a migratory and invasive phenotype [24]. Accordingly, we are the first in reporting PRAME’s positive influence on migration and invasion in the context of HCC.

Interference with MAPK signaling—such as blocking the receptor itself—modulated PRAME expression. Axl inhibition was capable of reducing PRAME expression within hours, while PRAME abundance remained constant for up to 24 h upon abrogating MAPK signaling. Blocking of Axl itself completely prevents activation of downstream cascades as well as potential heterodimerization with other RTKs, while MAPK inhibitors specifically interfere with one downstream effector pathway [10]. Fitting with our observation of Gas6/Axl/Mek/ERK1/2-induced PRAME expression, analysis of the PRAME locus revealed the presence of a binding motif for TEAD1 (https://motifmap.ics.uci.edu/), which can be activated in a MAPK-dependent manner [59]. Additionally, the PRAME locus harbors motifs for LEF1 (https://motifmap.ics.uci.edu/), which associates with an EMT-favoring signature in Axl-expressing settings (Figure 2A) and positively associates with PRAME (Appendix A). Even though we did not observe increased activation of Src, Akt or PLCγ upon Gas6 stimulation, we cannot rule out that Axl activates other pathways in a Gas6-independent fashion [10], potentially contributing to PRAME induction. Furthermore, we hypothesize that PRAME can be induced via an Axl/MAPK-independent mechanism, which is supported by the 46% of PRAME-positive, yet Axl-negative patient samples.

Functions of PRAME beyond repressing RA signaling in HCC are very likely as genetic interference with PRAME does not alter reporter gene activity stimulated by administration of all-trans retinoic acid [41]. Thus, we performed IP-MS analysis to identify the PRAME interactome in the context of HCC cells which, as expected, did not detect retinoic acid receptors as binding partners. IP-MS revealed AGAP1, CCAR1 and SH3PXD2B as interaction partners among others. Interestingly, all three proteins are linked to tumor progression in various cancer entities. CCAR1, a transcriptional co-regulator, favors tumor cell proliferation and migration [60]. In HCC, the interaction of CCAR1 with β-catenin induces cancer stem cell enrichment via activation of Wnt target genes [61]. The same mechanism is utilized in 5-fluorouracil resistant colorectal cancer models as CCAR1 associates with β-catenin and triggers a stemness-like phenotype by inducing Myc among others [62]. Knockdown of SH3PXD2B reduces invadopodia formation and subsequently lung metastasis in in vivo models [63]. Recently, a genome-wide association study found links between AGAP1 variants and response to the anti-VEGF antibody Bevacizumab [64], which is used as a first-line treatment for HCC in combination with the anti-PD-L1 antibody Atezolizumab [2]. In this line, we found that *AGAP1*, *CCAR1* and *SH3PXD2B* are significantly higher expressed in HCC compared to normal tissue and (i) positively correlate with *PRAME* expression in tumor tissue of available HCC patient data and (ii) high expression of each interaction partner associates with poorer survival probability. In future studies, we aim to mechanistically address how PRAME exerts pro-oncogenic functions together with its interaction partners. ChIP- and ATAC-Seq will be performed to identify (i) PRAME-connected chromatin regions, (ii) how PRAME influences chromatin accessibility and (iii) the impact of PRAME on EMT, invasion and stemness signatures.

PRAME harbors multiple murine orthologues that exert different physiological functions in the context of stem cell maintenance, differentiation and spermatogenesis [65,66]. As the murine PRAME-like family is currently mainly studied in the context of developmental biology, it remains unclear which of them are upregulated in the context of cancer and whether they are involved in tumor development. Therefore, the generation of an adequate autochthonous mouse model to study their contribution to HCC is currently hardly accessible. Thus, we focused on the analysis of human HCC patient data rather than on generating PRAME-like KO mice. In accordance with previous studies [6,41], we found a link between PRAME and Axl expression and an unfavorable prognosis for HCC patients. In agreement with the in vitro analysis, we found increased expression of PRAME in Axl-stratified patient samples. PRAME expression increases with grading and stage of the primary tumor. Additionally, higher PRAME abundance associates with an invasive phenotype, together highlighting PRAME’s connection to disease progression.

Observation of the current study supports the hypothesis that the Gas6/Axl/ERK1/2/PRAME axis promotes HCC. Importantly, we focused on tumor cell-intrinsic Axl signaling; however, the receptor is also expressed on other cell types such as immune or endothelial cells, conferring anti-inflammatory capacities, potentially complicating anti-Axl treatment. Upon triggering inflammation via AOM-DSS, a significantly higher tumor burden was observed in Axl/Mer-deficient mice due to impaired response to over boarding inflammation [67]. In contrast, Axl is triggering liver fibrosis, which is the main risk factor for development of HCC [68]. Upon liver transplantation, Axl signaling is hepato-protective by counteracting hepatic injury by reducing pro-inflammatory signaling [69]. Thus, systemic Axl therapy not only targets malignant cells but also stromal and immune cells, thereby increasing the risk of potential unwanted yet manageable side effects, as reported recently [70]. Interfering with pro-oncogenic targets of tumor-intrinsic Axl signaling could serve as an alternative for those suffering from severe side effects from anti-Axl therapy. Moreover, the 46% of Axl-negative, yet PRAME-positive patients could benefit from targeting PRAME independently from intervention with Axl. Another important aspect is PRAME’s potential for CAR-T-cell therapy due to PRAME’s restricted expression [19,20]. Anti-tumor efficacy of PRAME-specific T cell clones was already suggested more than a decade ago in the context of graft-versus-host-disease [17]. PRAME-specific T-cells were efficient in recognizing tumor cell lines expressing the antigen while being incapable of recognizing non-malignant PRAME-negative cells such as hepatocytes [17]. However, as PRAME is frequently only expressed intracellularly rather than on the surface, the implementation of PRAME-specific T-cells is limited. Yet, recent advances in targeting non-surface tumor antigens lead to the emergence of T cell receptor mimics (^TCRm^) CAR-T-cells, capable of recognizing peptide/MHC complexes derived from tumor-antigens [71]. First results from acute myeloid leukemia models expressing intracellular PRAME with ^TCRm^CAR-T-cells showed strong anti-tumor efficacy in vitro and in vivo [18]—yet their effect on solid PRAME-positive cancers remains to be determined.

## 5. Conclusions

This study provides insights into the molecular aspects of Axl-driven HCC progression and identified the CTA PRAME as a tumor-promoting target gene of Axl. Its limited expression in healthy adults marks PRAME as an attractive therapeutic target with probably manageable adverse effects. The importance of PRAME for mesenchymal stem/stromal cell differentiation could potentially lead to off-target effects [57,72]. Further advances in the clinical development of PRAME-specific ^TCRm^CAR-T-cells could serve as a novel approach to target PRAME-positive tumors including HCC.

## Figures and Tables

**Figure 1 cancers-15-02415-f001:**
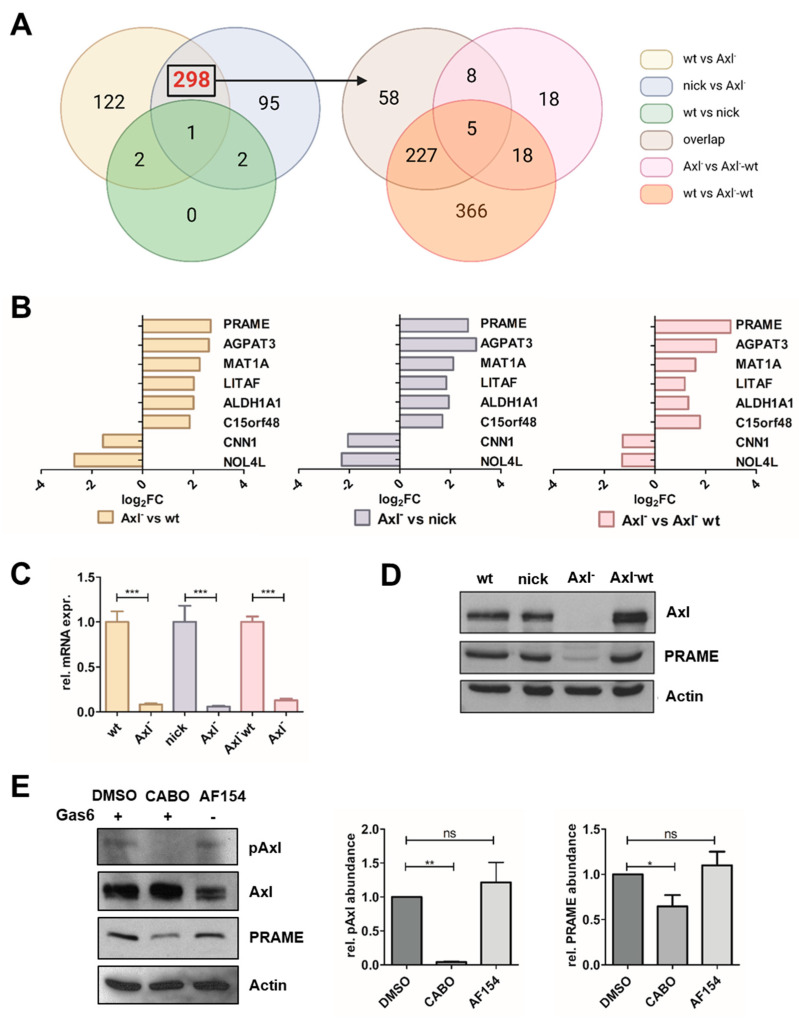
Identification of Gas6/Axl-regulated transcripts in HCC cells. (**A**) Axl target genes were identified by RNA-seq of HCC cells stimulated with 500 ng/mL Gas6 for 24 h. *p* < 0.05 and an absolute log_2_-fold change of at least +1.0 were used for selection of targets. Axl dependencies were defined by comparisons of Axl-positive (wt, nick) versus Axl-negative (Axl^−^) cells (left panel). As indicated by the arrow, 298 genes were further intersected with differentially expressed genes of Axl-reconstituted cells (Axl^−^-wt). (**B**) Log_2_-fold changes (Log_2_FCs) of 8 Axl targets in Axl^−^ vs. wt (yellow), Axl^−^ vs. nick (blue) and Axl^−^ vs. Axl^−^-wt (red). (**C**,**D**) Expression of PRAME transcript and protein levels as determined by qPCR and Western blotting, respectively. The expression of PRAME in Axl-expressing HCC cells was set to a value of 1. The expression of actin is shown as a loading control. (**E**) PRAME expression upon stimulation with 500 ng/mL Gas6 for 2 h in the absence (DMSO) or presence of 10 µM Axl inhibitor Cabozantinib (CABO) and upon administration of 900 ng/mL stimulatory Axl antibody (AF154) for 2 h. Protein levels were quantified by densitometry. wt, HLF; nick, HLF-nickase; Axl^−^, HLF-Axl-KO; Axl^−^-wt, HLF-Axl-KO-wt-Axl. Data are expressed as mean ± SD. ns: *p* > 0.05; *: *p* ≤ 0.05; **: *p* ≤ 0.01; ***: *p* ≤ 0.001.

**Figure 2 cancers-15-02415-f002:**
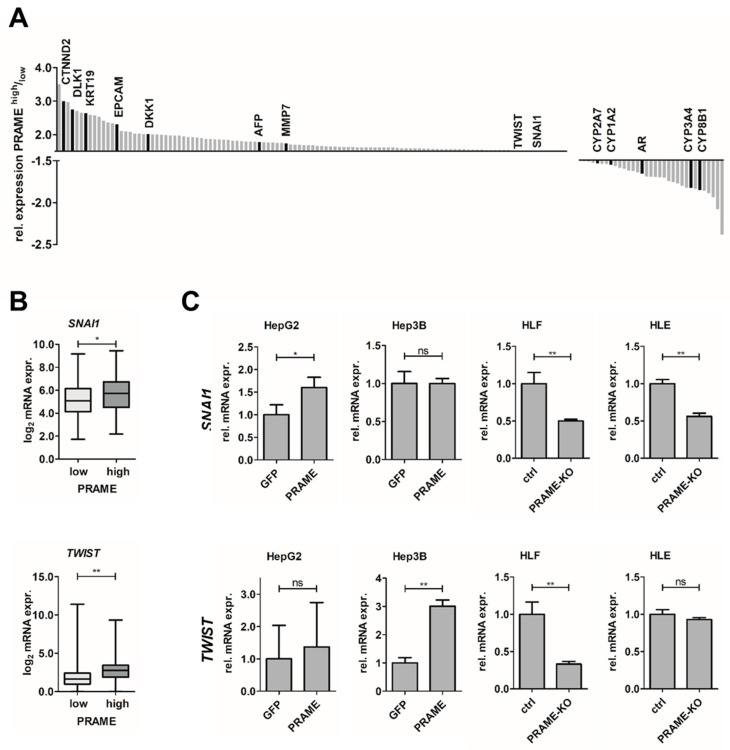
Expression of PRAME associates with EMT. (**A**) Summary of differentially expressed genes in PRAME-stratified HCC patients in publicly available HCC samples from TCGA. *n* = 372. (**B**) Expression of the EMT-TFs *SNAI1* (top) and *TWIST* (bottom) in *PRAME*−stratified HCC patient data from TCGA LIHC. *n* = 90 per group. (**C**) Modulation of EMT−TFs in PRAME-expressing epithelial cells (HepG2, Hep3B) and in mesenchymal cells (HLF, HLE) lacking PRAME expression (PRAME-KO). The expression of *SNAI1* and *TWIST* in GFP-expressing HCC cells (HepG2, Hep3B) and control HLF/HLE cells (ctrl) were set to a value of 1. Data are expressed as mean ± SD. ns: *p* > 0.05; *: *p* ≤ 0.05; **: *p* ≤ 0.01.

**Figure 3 cancers-15-02415-f003:**
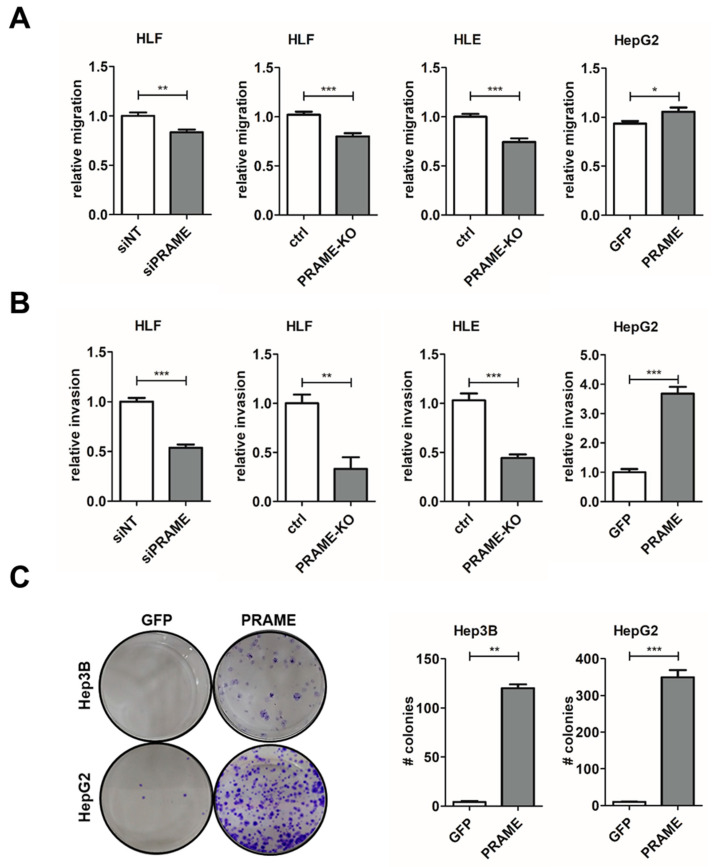
PRAME raises migratory and invasive capabilities of HCC cells. (**A**) Cell migration of Gas6-stimulated HLF cells upon PRAME knockdown (siPRAME), PRAME-KO cells (HLF, HLE) and of PRAME-expressing HepG2 cells as analyzed by wound healing assays. (**B**) Invasion of HCC cells as described in (**A**) after spheroid formation and inclusion into collagen gels. The migration (**A**) and invasion (**B**) of non-target siRNA (siNT) treated cells, GFP-expressing HCC cells (HepG2) and control HLF/HLE cells (ctrl) were set to a value of 1. (**C**) Clonogenic growth behavior (left panels) of Hep3B-PRAME/Hep3B-GFP (top) and HepG2-PRAME/HepG2-GFP cells (bottom) and its quantification (right panels). Data are expressed as mean ± SD. *: *p* ≤ 0.05; **: *p* ≤ 0.01; ***: *p* ≤ 0.001.

**Figure 4 cancers-15-02415-f004:**
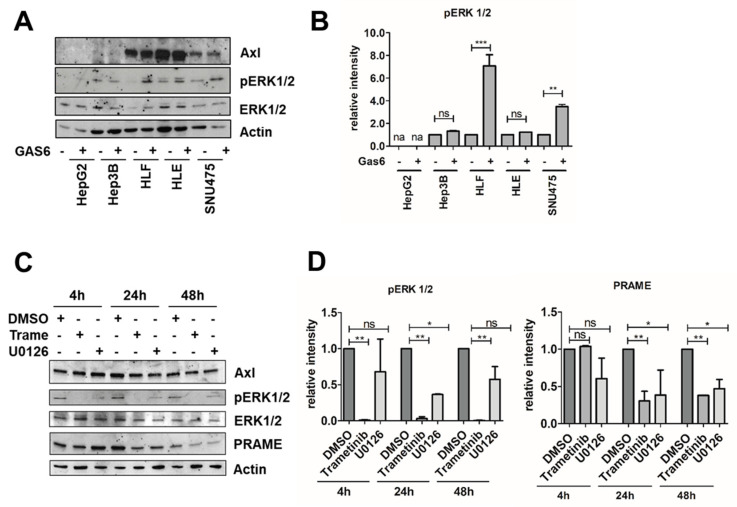
Gas6/Axl induces MAPK signaling and PRAME expression. (**A**) Levels of phosho−ERK1/2 (pERK1/2) were assessed by Western blotting upon stimulation of HepG2, Hep3B, HLF, HLE and SNU475 cells with 500 ng/mL Gas6 for 15 min. (**B**) Quantification of pERK1/2 levels by densitometry (na: not applicable). The abundance of pERK1/2 in the absence of Gas6 was set to a value of 1 for each liver cancer cell line. (**C**) PRAME expression together with pERK1/2 and total ERK1/2 levels in HLF cells were assessed by Western blotting upon administration of 1 µM of either MEK inhibitors Trametinib or U0126. The expression of actin is shown as loading control (**A**,**C**). (**D**) pERK1/2 and PRAME expression were quantified by densitometry. The expression of DMSO−treated control cells was set to a value of 1. Data are expressed as mean ± SD. ns: *p* > 0.05; *: *p* ≤ 0.05; **: *p* ≤ 0.01; ***: *p* ≤ 0.001.

**Figure 5 cancers-15-02415-f005:**
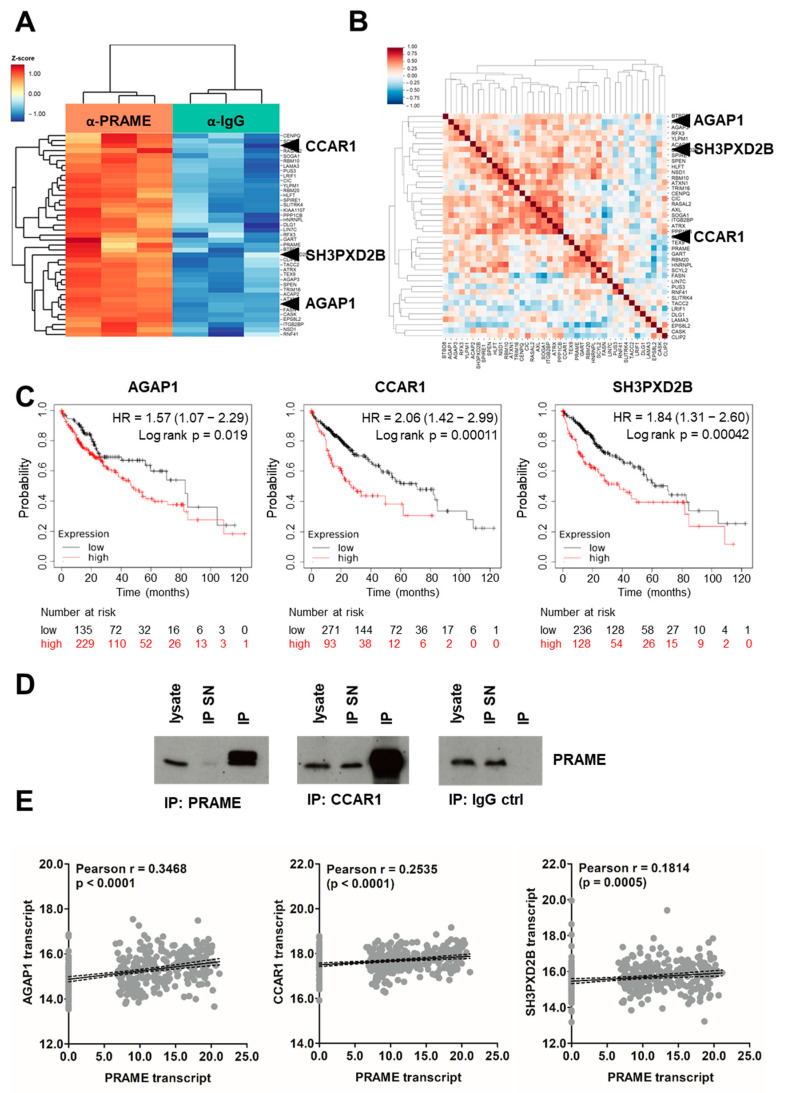
Immunoprecipitation reveals binding partners of PRAME. (**A**) Significantly enriched genes in anti−PRAME sample (orange) vs. IgG control (green) of anti−PRAME−IP−MS. (**B**) Positively and negatively correlated binding partners of PRAME in liver cancer cell lines using the cancer cell line encyclopedia. Higher resolution versions of (**A**,**B**) are included in the Appendix A. (**C**) Impact of the IP−MS-identified PRAME binding partners AGAP1, CCAR1 and SH3PXD2B on patient prognosis. AGAP1^high^, *n* = 229; AGAP1^low^, *n* = 135; CCAR1^high^, *n* = 93; CCAR1^low^, *n* = 271; SH3PXD2B^high^, *n* = 128; SH3PXD2B^low^, *n* = 236. (**D**) IP performed with anti-PRAME, anti-CCAR1 and IgG control antibody and subsequent assessment of PRAME expression via Western blotting. SN, supernatant. (**E**) Correlations between the expression of PRAME and the IP−MS−identified binding partners AGAP1, CCAR1 and SH3PXD2B in primary tumor samples using the publicly available patient dataset TCGA LIHC (*n* = 372). Each grey dot represents one sample. Regression lines are shown in solid black and the 95% confidence intervals as dashed black lines.

**Figure 6 cancers-15-02415-f006:**
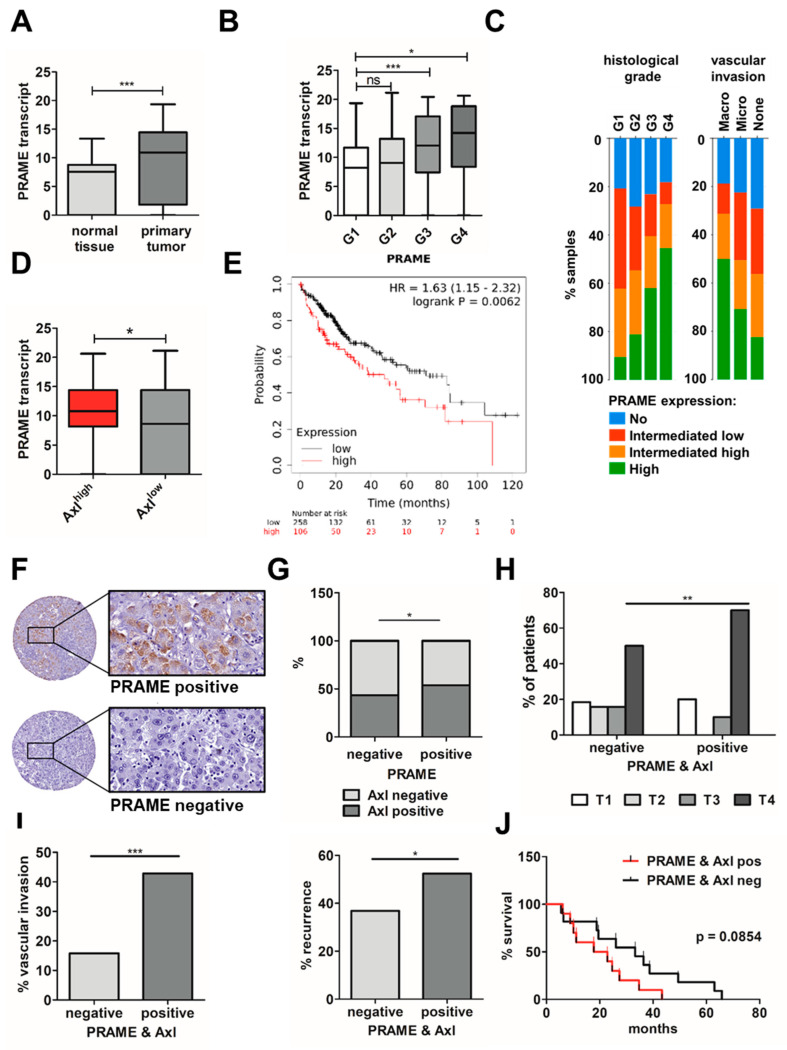
Correlation of PRAME and Axl expression levels with prognosis and survival of HCC patients. (**A**) PRAME transcript levels in normal liver tissue and primary HCC (*n* = 52) and (**B**) stratified according to different histological grades (G1–G4) from publicly available TCGA LIHC data. G1, *n* = 55; G2, *n* = 177, G3, *n* = 122, G4, *n* = 12. (**C**) Relative PRAME expression in percentage per histological grades (G1–G4; *p*-value: 1.839 × 10^−4^) and its contribution to invasiveness (*p*-value: 0.0429) in publicly available HCC patient samples using cBioPortal. *n* = 93 per group. (**D**) PRAME expression in Axl-stratified TCGA LIHC data. *n* = 93 per group. (**E**) Combined effect of PRAME and Axl on patient survival (red: PRAME^high^ + Axl^high^, *n* = 106; black: PRAME^low^ + Axl^low^, *n* = 258). (**F**) Representative images from immunohistochemical analysis of PRAME-positive and PRAME-negative HCC tissue samples. Rectangles represent a 30-fold magnification. (**G**) Distribution of Axl expression in PRAME-positive and -negative samples in percent. PRAME/Axl-negative, *n* = 38; PRAME/Axl-positive, *n* = 20; PRAME-positive, Axl-negative, *n* = 18; PRAME-negative, Axl-positive, *n* = 29. (**H**) Distribution of PRAME-positive samples per tumor stage. PRAME/Axl-negative: T1, *n* = 7, T2, *n* = 6, T3, *n* = 6, T4, *n* = 19; PRAME/Axl-positive: T1, *n* = 4, T2, *n* = 0, T3, *n* = 2, T4, *n* = 14, Tnd, *n* = 1. (**I**) Vascular invasion (left) and recurrence (right) in PRAME/Axl-positive and PRAME/Axl-negative HCC samples. PRAME/Axl-negative: vascular invasion, yes, *n* = 6, no, *n* = 32, recurrence, yes, *n* = 14, no, *n* = 24; PRAME/Axl-positive: vascular invasion, yes, *n* = 9, no, *n* = 11, recurrence, yes, *n* = 10, no, *n* = 10. (**J**) Combined effect of PRAME and Axl protein levels on patient survival (black: PRAME/Axl-negative, *n* = 38; red: PRAME/Axl-positive, *n* = 20). Data are expressed as mean ± SD. ns: *p* > 0.05; *: *p* ≤ 0.05; **: *p* ≤ 0.01; ***: *p* ≤ 0.001.

## Data Availability

Datasets used for bioinformatic analysis are available in cBioportal (https://www.cbioportal.org/), in the Xenabrowser (https://xenabrowser.net/) and in the (https://kmplot.com) for hepatocellular carcinoma (LIHC), The Cancer Genome Atlas. The human tissue microarray data are not publicly available due to privacy and ethical restrictions.

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
