# Peer review of "PRAME Is a Novel Target of Tumor-Intrinsic Gas6/Axl Activation and Promotes Cancer Cell Invasion in Hepatocellular Carcinoma"

_cancers, 2023, doi:10.3390/cancers15092415_

Round 1

Reviewer 1 Report

The authors showed that PRAME induces hepatic dedifferentiation, epithelial-to-mesenchymal transition and invasiveness in liver cancer cells. Together, these data provide evidence that PRAME is induced by the Gas6/Axl/Mek/Erk1/2 signaling axis and exerts pro-oncogenic functions in HCC. The HCC cell specific upregulation and antigenicity suggest PRAME as a target for CAR-T cell therapy.

My comments:

The authors were ambitious to convince the audience to believe their theory. But it was too complicated with so many genes mixed together and difficult to be understood. In addition, the conclusion was also not convincing by their methodology.

Author Response

Dear Prof. Mok,

Many thanks for your efficient handling of the manuscript ID: cancers-2333836 entitled “PRAME is a novel target of tumor-intrinsic Gas6/Axl activation and promotes cancer cell invasion in hepatocellular carcinoma” by Viola Hedrich, Kristina Breitenecker, Gregor Ortmayr, Franziska Pupp, Heidemarie Huber, Doris Chen, Sarthak Sahoo, Mohit Kumar Jolly, and myself, which we herewith submit again in revised form for publication as an original research article in “Cancers”.

We would like to thank the editorial for the very useful comments and the effort so far. We have closely followed the suggestions of the reviewers, and have introduced corresponding changes into the text. For your convenience and to specify the revisions in the manuscript more clearly, the changes are marked up using the “Track Change” mode.

In the following, please find a detailed point-by-point discussion of the changes introduced into the text and comments to the issues raised by the referees.

Notes to the comments of reviewer 1

To the comment:

The authors were ambitious to convince the audience to believe their theory. But it was too complicated with so many genes mixed together and difficult to be understood. In addition, the conclusion was also not convincing by their methodology.

Thank you for this suggestion which we closely followed. We particularly attempted to improve introduction, results and conclusions. (i) Accordingly, we removed CAR-T cells from the simple summary and key words in order to avoid confusion of the readership. (ii) We edited the introduction regarding PRAME and focused specifically on its contribution in malignant settings. We briefly stated the pro-oncogenic function of PRAME in other cancer entities (page 3, line 73-88). (iii) We cross-checked references and modified them, resulting in the inclusion of additional 15 references. (iv) We included more details regarding the methodology of identifying Axl targets (page 6, line 204-207). (v) For a straight forward presentation in the Results section, we moved Figure 2A into the supplement (now Figure S1B). As a consequence, we arranged the former Figure S1B – S1E as an independent supplemental Figure S2A – S2D. All other supplemental figures were left unchanged except that they all have a new number (Figure S3 à Figure S4; Figure S4 à Figure S5; Figure S5 à Figure S6; Figure S6 à Figure S7). The cross references in the manuscript were changed accordingly. Furthermore, for a better understanding and for easier traceability, we explained in more detail why we focused on the IP-MS-identified PRAME binding partners AGAP1, CCAR1 and SH3PXD2B. In accordance, we rearranged Figure 5C-E and the corresponding figure legend. We rearranged passages that were written in a possibly confusing way in 3.2. and 3.5. analysis (page 11f, line 366-387; page 13, line 404-419; page 20, line 527-538). (vi) We added an intermediate conclusion in 3.2. after describing the in silico analysis (page 11, line 386-387), and modified conclusion statements in 3.3. (page 15, line 450-453). (vi) An native speaking working in cancer biology polished English.

Notes to the comments of reviewer 2

To comment 1:

The authors showed the significant relation between the PRAME expression and EMT-related protein expressions (TWIST and SNAI1). In addition, the authors suggested the significant role of PRAME in the dedifferentiation of HCC cells. The reviewer would like to ask the morphological phenotype of HCC cells. Was any morphological change detected by the overexpression or knockdown/knockout of PRAME?

Thank you for the comment. We did not observe any obvious changes in PRAME-modified HCC cells at the level of morphology. A corresponding notion was added in the section 3.2. (page 14, line 432-434).

To comment 2:

In Fig. 4A and 4B, there are small problems. In Fig, 4B, the word ‘GAS6’ should be displayed.

Thank you for your particular attention. Of course, we have incorporated “Gas6” in Figure 4B.

To comment 3:

In Fig. 6, there are no information regarding the number of patients in each group (Ex. G1-G4, macro/micro/none vascular invasion, T1-T4 etc.). The reviewer would request to display the number of patients in each figure or create new table to display the information of analyzed patients.

Thank you for this request. As recommended by the reviewer, the numbers of patients were included into the legend of Figure 6. Additionally, the numbers of patients were included into the legends of Figure 2A, Figure 2B, the updated Figures 5C and 5E, as well as the updated Figure S7.

To comment 4:

According to Fig. 6H, approx. 50% of patients with PRAME&Axl-negative were categorized in T4. However, in Fig. 6I, only approx. 15% patients with PRAME&Axl-negative showed vascular invasion. In pathological criteria, the patients categorized in T4 show severe invasion of cancer cells to portal vein or hepatic vein. Therefore, the reviewer cannot understand the reason why the percentage of patients with vascular invasion is so low despite the large percentage of patients with T4. The reviewer would request the authors to explain it.

We appreciate this request. Notably, we assessed the expression of PRAME and Axl in a cohort of patients that all received orthotopic liver transplantation (OLT), a treatment option for early HCC patients. The majority of patients was in the stage of T4 (92/133), however, only a small subset of these patients showed invasion (4/92), and very few lymph node presence and metastasis (2/92) according to the evaluation of pathologists (which made them suitable for transplantation). Unfortunately, the pathologists are unavailable for further consultation to scrutinize why such a high number of T4 stages were selected for OLT. We hope for the understanding of the reviewer that we do not to include corresponding statements into the text in order to avoid confusion of the readership. However, we are willing to rearrange the text upon the particular request of the reviewer.

Notes to the comments of reviewer 3

To the comment:

According to the academic guidelines, the raw data of RNA-seq should be uploaded to the GEO database, and the accession number could be included in the 2.7 section.

Thank you for this suggestion. We have uploaded the raw data of RNA-seq to the GEO database. A corresponding notion mentioning the accession number will be included into the section 2.7. of “Materials and Methods” (page 6, line 194).

Overall, we have responded to all issues raised by the reviewers and we believe that we could sufficiently answer all requests. With the inclusion of suggested intext revisions, we are convinced that the manuscript is significantly improved. We hope that you find our revised work suitable for publication in ”Cancers”.

Looking forward to your evaluation, I remain

Yours sincerely

Wolfgang Mikulits, Ph.D.

Reviewer 2 Report

The present study investigated the oncological role of PRAME expression in hepatocellular carcinoma (HCC). The expression of PRAME induced the proliferation, migration and invasion of HCC cells. The elevated expression of PRAME was also related to the induction of EMT. The inhibition of MEK induced the reduced expression of PRAME as well as pERK1/2. AGAP1, CCAR1 and SH3PXD2B were identified as the binding partners of PRAME. Elevated expression of PRAME was also detected in the cancer tissues of HCC patients and related to the worse cancer behavior and patients’ prognosis. The reviewer considers that the present study showed new evidences of the molecular oncology regarding PRAME. Although there are sufficient contents in the present manuscript, the reviewer would like to ask some queries to the authors as described below.

1.     The authors showed the significant relation between the PRAME expression and EMT-related protein expressions (TWIST and SNAI1). In addition, the authors suggested the significant role of PRAME in the dedifferentiation of HCC cells. The reviewer would like to ask the morphological phenotype of HCC cells. Was any morphological change detected by the overexpression or knockdown/knockout of PRAME?

2.     In Fig. 4A and 4B, there are small problems. In Fig, 4B, the word ‘GAS6’ should be displayed.

3.     In Fig. 6, there are no information regarding the number of patients in each group (Ex. G1-G4, macro/micro/none vascular invasion, T1-T4 etc.). The reviewer would request to display the number of patients in each figure or create new table to display the information of analyzed patients.

4.     According to Fig. 6H, approx. 50% of patients with PRAME&Axl-negative were categorized in T4. However, in Fig. 6I, only approx. 15% patients with PRAME&Axl-negative showed vascular invasion. In pathological criteria, the patients categorized in T4 show severe invasion of cancer cells to portal vein or hepatic vein. Therefore, the reviewer can not understand the reason why the percentage of patients with vascular invasion is so low despite the large percentage of patients with T4. The reviewer would request the authors to explain it.

Author Response

(The authors gave the same response as above.)

Reviewer 3 Report

This manuscript reported PRAME is a novel target of HCC, which could promote the tumor progression. This manuscript was well prepared, and the data also support the conclusion. I recommend the acceptance of this manuscript after some minor revision.

1. According to the academic guidelines, the raw data of RNA-seq should be uploaded to GEO database, and the accession number could be included in the 2.7 section.

Author Response

Dear Prof. Mok,

Many thanks for your efficient handling of the manuscript ID: cancers-2333836 entitled “PRAME is a novel target of tumor-intrinsic Gas6/Axl activation and promotes cancer cell invasion in hepatocellular carcinoma” by Viola Hedrich, Kristina Breitenecker, Gregor Ortmayr, Franziska Pupp, Heidemarie Huber, Doris Chen, Sarthak Sahoo, Mohit Kumar Jolly, and myself, which we herewith submit again in revised form for publication as an original research article in “Cancers”.

We would like to thank the editorial for the very useful comments and the effort so far. We have closely followed the suggestions of the reviewers, and have introduced corresponding changes into the text. For your convenience and to specify the revisions in the manuscript more clearly, the changes are marked up using the “Track Change” mode.

In the following, please find a detailed point-by-point discussion of the changes introduced into the text and comments to the issues raised by the referees.

Notes to the comments of reviewer 1

To the comment:

The authors were ambitious to convince the audience to believe their theory. But it was too complicated with so many genes mixed together and difficult to be understood. In addition, the conclusion was also not convincing by their methodology.

Thank you for this suggestion which we closely followed. We particularly attempted to improve introduction, results and conclusions. (i) Accordingly, we removed CAR-T cells from the simple summary and key words in order to avoid confusion of the readership. (ii) We edited the introduction regarding PRAME and focused specifically on its contribution in malignant settings. We briefly stated the pro-oncogenic function of PRAME in other cancer entities (page 3, line 73-88). (iii) We cross-checked references and modified them, resulting in the inclusion of additional 15 references. (iv) We included more details regarding the methodology of identifying Axl targets (page 6, line 204-207). (v) For a straight forward presentation in the Results section, we moved Figure 2A into the supplement (now Figure S1B). As a consequence, we arranged the former Figure S1B – S1E as an independent supplemental Figure S2A – S2D. All other supplemental figures were left unchanged except that they all have a new number (Figure S3 à Figure S4; Figure S4 à Figure S5; Figure S5 à Figure S6; Figure S6 à Figure S7). The cross references in the manuscript were changed accordingly. Furthermore, for a better understanding and for easier traceability, we explained in more detail why we focused on the IP-MS-identified PRAME binding partners AGAP1, CCAR1 and SH3PXD2B. In accordance, we rearranged Figure 5C-E and the corresponding figure legend. We rearranged passages that were written in a possibly confusing way in 3.2. and 3.5. analysis (page 11f, line 366-387; page 13, line 404-419; page 20, line 527-538). (vi) We added an intermediate conclusion in 3.2. after describing the in silico analysis (page 11, line 386-387), and modified conclusion statements in 3.3. (page 15, line 450-453). (vi) An native speaking working in cancer biology polished English.

Notes to the comments of reviewer 2

To comment 1:

The authors showed the significant relation between the PRAME expression and EMT-related protein expressions (TWIST and SNAI1). In addition, the authors suggested the significant role of PRAME in the dedifferentiation of HCC cells. The reviewer would like to ask the morphological phenotype of HCC cells. Was any morphological change detected by the overexpression or knockdown/knockout of PRAME?

Thank you for the comment. We did not observe any obvious changes in PRAME-modified HCC cells at the level of morphology. A corresponding notion was added in the section 3.2. (page 14, line 432-434).

To comment 2:

In Fig. 4A and 4B, there are small problems. In Fig, 4B, the word ‘GAS6’ should be displayed.

Thank you for your particular attention. Of course, we have incorporated “Gas6” in Figure 4B.

To comment 3:

In Fig. 6, there are no information regarding the number of patients in each group (Ex. G1-G4, macro/micro/none vascular invasion, T1-T4 etc.). The reviewer would request to display the number of patients in each figure or create new table to display the information of analyzed patients.

Thank you for this request. As recommended by the reviewer, the numbers of patients were included into the legend of Figure 6. Additionally, the numbers of patients were included into the legends of Figure 2A, Figure 2B, the updated Figures 5C and 5E, as well as the updated Figure S7.

To comment 4:

According to Fig. 6H, approx. 50% of patients with PRAME&Axl-negative were categorized in T4. However, in Fig. 6I, only approx. 15% patients with PRAME&Axl-negative showed vascular invasion. In pathological criteria, the patients categorized in T4 show severe invasion of cancer cells to portal vein or hepatic vein. Therefore, the reviewer cannot understand the reason why the percentage of patients with vascular invasion is so low despite the large percentage of patients with T4. The reviewer would request the authors to explain it.

We appreciate this request. Notably, we assessed the expression of PRAME and Axl in a cohort of patients that all received orthotopic liver transplantation (OLT), a treatment option for early HCC patients. The majority of patients was in the stage of T4 (92/133), however, only a small subset of these patients showed invasion (4/92), and very few lymph node presence and metastasis (2/92) according to the evaluation of pathologists (which made them suitable for transplantation). Unfortunately, the pathologists are unavailable for further consultation to scrutinize why such a high number of T4 stages were selected for OLT. We hope for the understanding of the reviewer that we do not to include corresponding statements into the text in order to avoid confusion of the readership. However, we are willing to rearrange the text upon the particular request of the reviewer.

Notes to the comments of reviewer 3

To the comment:

According to the academic guidelines, the raw data of RNA-seq should be uploaded to the GEO database, and the accession number could be included in the 2.7 section.

Thank you for this suggestion. We have uploaded the raw data of RNA-seq to the GEO database. A corresponding notion mentioning the accession number will be included into the section 2.7. of “Materials and Methods” (page 6, line 194).

Feeback by GEO:

"[sent to: "mikulitscalendar@gmail.com"]

Thank you for using the GEO Submission form.

Transferred files are placed into the processing queue and will be reviewed within 5 business days; expect to receive an email from GEO curators with your GEO accession numbers, or questions about your submission. We can be contacted at geo@ncbi.nlm.nih.gov if you do not hear from us within the allotted time, or if you require additional assistance."

Overall, we have responded to all issues raised by the reviewers and we believe that we could sufficiently answer all requests. With the inclusion of suggested intext revisions, we are convinced that the manuscript is significantly improved. We hope that you find our revised work suitable for publication in ”Cancers”.

Looking forward to your evaluation, I remain

Yours sincerely

Wolfgang Mikulits, Ph.D.

Round 2

Reviewer 1 Report

None.